# Immunocytochemical Labelling of Haematological Samples Using Monoclonal Antibodies

**DOI:** 10.3390/cells11010127

**Published:** 2021-12-31

**Authors:** Wendy N. Erber

**Affiliations:** School of Biomedical Sciences, The University of Western Australia, Perth, WA 6009, Australia; wendy.erber@uwa.edu.au

**Keywords:** monoclonal antibodies, diagnostics, immunoenzymatic

## Abstract

I reflect on my experience working with David Y. Mason in the Leukaemia Research Laboratories in the Nuffield Department of Pathology at the University of Oxford in the early 1980s. This was soon after the first monoclonal antibodies had been produced, which led to an exciting and productive time in biological discovery and pathology diagnostics. A specific focus in the laboratory was the development of immunoenzymatic staining methods that would enable monoclonal antibodies to be applied in diagnostic practice. This paper describes the work that led to the performance of immuno-alkaline phosphatase staining on blood and bone marrow smears, the success of which changed leukaemia diagnosis.

## 1. Introduction

The key to the effective management of haematological malignancies is accurate diagnosis and classification. Cellular morphology had been the mainstay of diagnosis since Ehrlich developed staining methods in 1891. This was supplemented by cytochemical staining techniques to identify specific patterns of reactivity based on the presence of cellular enzymes (e.g., peroxidase and acid phosphatase). Subsequently, polyclonal antisera and immunofluorescence was added to the diagnostic armamentarium. Immunofluorescent labelling was first developed by Albert H. Coons in 1941 and adopted for use in leukaemia diagnosis in the 1970s [1,2]. Commencing with well-characterized polyclonal antisera, to T cells, B cells, heavy and light chains of immunoglobulin, the common acute lymphoblastic leukaemia antigen, and terminal deoxynucleotidyl antigen (TdT), these methods could establish that a leukaemia was of T- or B-cell origin, or to be a non-T or non-B (common) acute lymphoblastic leukaemia [3,4]. This was primarily performed in research settings since immunofluorescence was not in the routine repertoire of diagnostic haematology laboratories.

Monoclonal antibodies began to be used to assist in leukaemia diagnosis soon after their first discovery by Köhler and Milstein in 1975 [5]. By the early 1980s, monoclonal antibodies were being used for leukaemia phenotyping by immunofluorescent microscopy and continued primarily to be undertaken in research laboratories [6,7]. This was an important breakthrough. Seeing the importance and relevance of the results obtained, haematologists wished to do testing within their own setting. However, immunofluorescence was outside the scope of normal haematology. The drawbacks included the need for fluorescent microscopes and the inability to visualize cells. Alternate techniques were required if immunological techniques took advantage of monoclonal antibody technology, which were to be used routinely in diagnostic laboratories.

## 2. Immuno-Enzymatic Staining of Cell Smears

David Mason was interested in developing techniques that would allow cell phenotyping to be performed on cell smears, the routine “tool” in diagnostic haematology. He commenced work in this area in the mid-1970s, exploring the detection of cellular antigens on blood smears using an immunoperoxidase staining reaction [8]. He was able to demonstrate that cellular antigens survived smearing and that morphology and the antigenic stain could be detected simultaneously on individual cells. Although a breakthrough, endogenous peroxidase within many blood and bone marrow cell (i.e., erythrocytes and myeloperoxidase in myeloid cells) could not be totally inhibited and obscured the specific cellular antigen staining. Immunoperoxidase staining was, therefore, not really suitable for diagnostic application to routine haematological samples [9]. An alternate enzyme label was required, and alkaline phosphatase became the way forward using the alkaline phosphatase anti-alkaline phosphatase (APAAP) method [10]. Before taking this forward assessing cell smears required assurance that endogenous alkaline phosphatase, present within neutrophils, could be quenched without denaturing cellular antigens. The addition of levamisole, indeed, blocked neutrophil alkaline phosphatase without interfering with antigen expression. Levamisole specifically inhibited non-intestinal forms of alkaline phosphatase, but did not affect intestinal alkaline phosphatase, i.e., the enzyme form in APAAP (calf intestinal alkaline phosphatase).

The next step was to determine whether APAAP could be used with monoclonal antibodies on cell smears. The first study, by Moir et al. (1983), showed that the APAAP method and the immunofluorescent method gave identical results when blood and bone marrow samples were labelled [11]. This was performed on washed cytocentrifuged mononuclear cell preparations of leukaemia samples, and 16 different antigens were assessed. The success of this led to analysis of directly prepared air-dried smears of peripheral blood and bone marrow [12]. Extensive studies confirmed that the method was applicable to all types of directly prepared air-dried cell smears or cytocentrifuged preparations of blood, bone marrow, fine needle aspirates, or body fluids (e.g., cerebrospinal fluid or pleural fluid). We showed that smears could be retained at room temperature for up to 7 days without any loss of cellular antigens or stored at −20 °C indefinitely, without loss of immuno-reactivity. Cell fixation to stabilize the cell membrane was a crucial step prior to antigen labelling. Many fixatives were attempted, each with different effects on morphology and antigen expression. A compromise had to be reached between optimizing antigen expression without compromising morphology. Blocking endogenous alkaline phosphatase enzyme activity with levamisole was shown to be essential to avoid non-specific neutrophil staining. The most commonly used chromogenic substrate (Fast Red) gave a bright red signal for antigen-positive cells which contrasted with the blue haematoxylin nuclear counterstain. Together, this was sufficient to be able to identify individual cell types and assess their phenotype (Figure 1 and Figure 2). Standard light microscopy was used to assess antigen staining, and cells of interest were identified by their morphology and antigen expression by deposition of the red chromogenic substrate. There was no background staining of the serum or granulocytic cells.

## 3. Application of APAAP Staining for Cell Smears

The reliability of APAAP staining of cell smears was demonstrated on T- and B-cell populations in blood smears [12]. From there, APAAP was evaluated as a method to phenotype leukaemia. This was in a study of over 250 cases of haematological malignancy, utilizing an extensive panel of monoclonal antibodies [13]. This showed the method could identify lineage-associated and lineage-specific antigens, as well as those associated with stage of differentiation (e.g., CD34; TdT) (Figure 3 and Figure 4). Further, cellular morphology, the tool of the haematologist, was preserved so that the leukemic cell could be identified. A number of papers were published at this time which described the application of the method to a range of types of acute and chronic leukaemias as well as metastatic marrow infiltrates [14,15]. This work was deemed a major breakthrough for haematologists; the APAAP technique provided a simple method that could be used on routine specimens and could be incorporated into diagnostic haematology laboratories.

The APAAP staining method for cells smears had many attributes for clinical use, including: (a)Sample: only a small sample (smear of blood or bone marrow aspirate) was required for multiple antigen analyses;(b)Preparation: routinely prepared air-dried cell smears could be stained without any need for preliminary mononuclear cell separation;(c)Storage: air-dried smears could be freshly prepared, stained, or stored for later analysis. This had the added advantage that routine smears prepared without prior knowledge of the presence of leukaemia could be analysed;(d)Cellular localization of antigens: both surface membrane and intracellular (cytoplasmic, such as myeloperoxidase, and nuclear, such as TdT) antigens could be detected. This was particularly important to detect progenitor cells where lineage-associated molecules are only expressed within the cell early in maturation (e.g., cytoplasmic CD22 in B-cell development, and cytoplasmic CD3 in T-cells);(e)Compatibility with light microscopy: cells stained by APAAP retained their morphological detail. Morphologically abnormal cells could be identified, even when in small numbers, and then assessed for the antigen label. This simultaneous visualization of the cells and stain by conventional light microscopy was a huge benefit for diagnostic haematologists. Some scenarios were straightforward, e.g., an antigen expressed (i.e., positive staining) by an abnormal cell infiltrate, thereby giving the phenotype of the disease. In other situations when only a few abnormal cells were present in the sample, these could be identified by the cytological appearance and then assessed for antigen expression, e.g., large anaplastic cells or the cells of interest are present in clusters;(f)Rare events: since the APAAP labelling reaction product was red against a background of unstained cells (identified by nuclear haematoxylin alone), rare antigen-positive events stood out and were easily detected. Examples include the positive identification of neoplastic cells that are not visible on standard microscopy, e.g., CD30-positive Hodgkin cells or cytokeratin-positive epithelial cells, indicating marrow metastasis (Figure 3 and Figure 4). Estimates given of the sensitivity for APAAP detection of infrequent events were 1 positive cell in 50,000 (0.002%) [16];(g)Permanence of the APAAP stained preparations: once stained and mounted with a glass coverslip, the APAAP reaction product does not fade. Because the labelled preparation was permanent, there was opportunity for storage and subsequent review. Retrospective reviews and audits were possible with archival material as part of quality assurance exercises or for research projects;(h)No specialized equipment was required.

## 4. Discussion

The APAAP method, when applied to blood and bone marrow smears, therefore, had much to offer the diagnostic haematologist. Applications included enumeration of lymphocyte populations, classification of leukaemia, and detection of metastatic marrow infiltrates. It brought immunophenotyping into diagnostic haematology laboratories as it was a cheap and practical alternative to immunofluorescent staining to detect cell antigens. This proved to be an enormous advance for the diagnosis of blood and bone marrow diseases, and specifically leukaemias [17]. Leukaemia immunophenotyping moved out of research laboratories and into hospital laboratories. Patients were the beneficiaries as disease-appropriate management could be commenced without delay. Recommendations were made, and the APAAP method become the method of choice for leukaemia typing in diagnostic laboratories. Haematologists and laboratory scientists around the United Kingdom visited the Mason laboratory to learn the technique.

By the late 1980s, the APAAP method had gained such widespread acceptance as the preferred immunophenotyping technique for leukaemia diagnosis that quality assurance was required. The United Kingdom National External Quality Assessment Scheme (UK NEQAS) introduced a quality assurance program to monitor the performance of the method in hospital laboratories. The scheme operated in 12 countries, demonstrating the global impact of APAAP for staining leukaemia samples. This was acknowledged in 2018 at the 30th Anniversary Congress of the “UK NEQAS for Leucocyte Immunophenotyping”. Two decades after the initial description, it was shown that APAAP staining of cell smears could be performed using automated immunostainers. This gave greater standardization with benefits for quality and reproducibility whilst maximizing efficiency and reducing turn-around-time [18].

## 5. Conclusions

Immunophenotyping of haematological malignancies on directly prepared blood and bone marrow smears using immunoenzymatic staining methods, and particularly APAAP, was a major breakthrough. Although an “invention”, APAAP and the staining of blood and bone marrow smears was never patented. This was not David’s way; he was generous and collaborative, and wanted APAAP to be used as widely as possible so that patients with leukaemia and other malignancies would gain benefit. The APAAP technique is now rarely applied to cell smears having been largely replaced by automated high-throughput flow cytometric immunophenotyping. However, in the mid-1980s and 1990s, this simple and cheap APAAP method was a significant development which had revolutionized leukaemia diagnosis globally.

## Figures and Tables

**Figure 1 cells-11-00127-f001:**
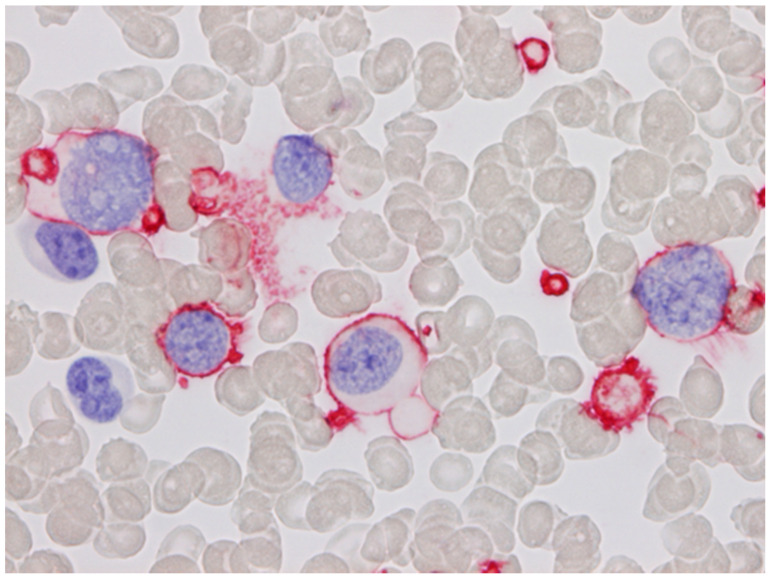
Acute megakaryoblastic leukaemia stained with CD61 using the APAAP method. The blast cells show surface staining. Platelets are also positive (blood smear; haematoxylin counterstain. ×1000).

**Figure 2 cells-11-00127-f002:**
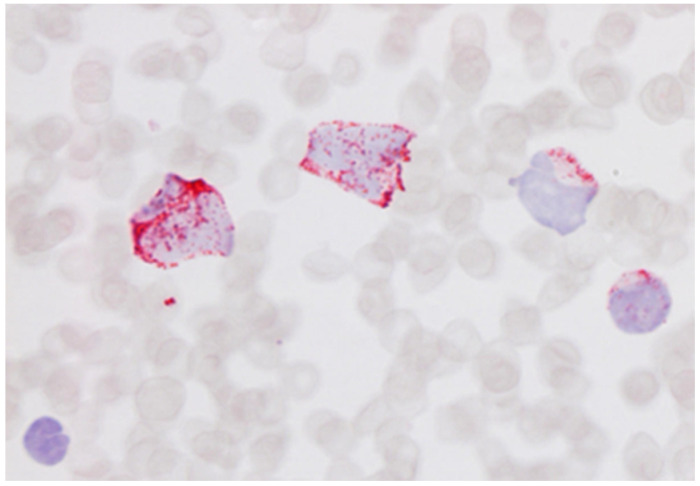
Blood film of acute myeloblastic leukaemia stained with an anti-myeloperoxidase antibody showing cytoplasmic positivity of the blast cells (blood smear; haematoxylin counterstain. ×1000).

**Figure 3 cells-11-00127-f003:**
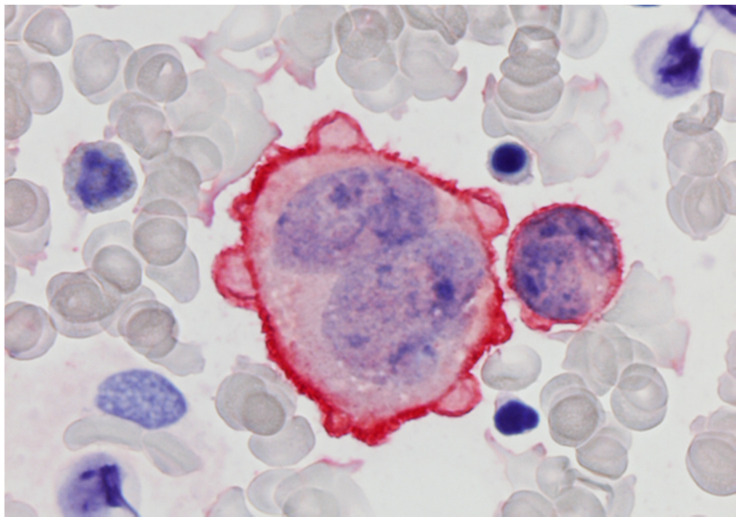
Anaplastic large cell lymphoma cells in bone marrow stained with CD30 using the APAAP method (bone marrow aspirate; haematoxylin counterstain. ×1000).

**Figure 4 cells-11-00127-f004:**
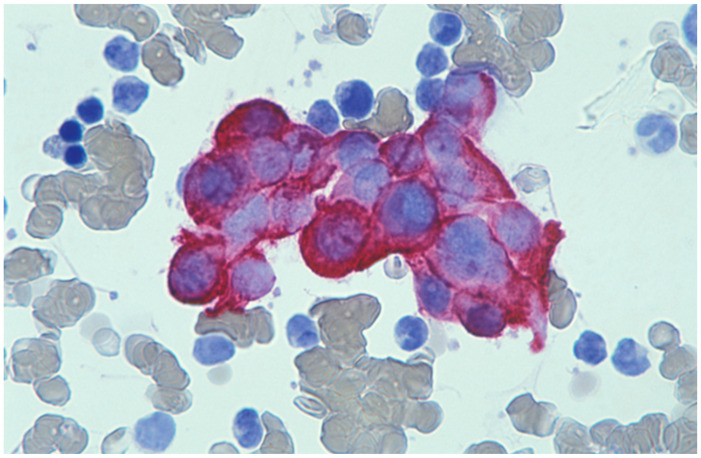
Bone marrow aspirate stained with a cytokeratin antibody (LP34) showing metastatic breast cancer cells (bone marrow aspirate; haematoxylin counterstain. ×1000).

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
