# Peer review of "Immunocytochemical Labelling of Haematological Samples Using Monoclonal Antibodies"

_cells, 2021, doi:10.3390/cells11010127_

Round 1

Reviewer 1 Report

Title: Immunocytochemical labelling of haematological samples using monoclonal antibodies

Dr. Wendy Erber shed light on her experience,  working with David Y. Mason in the Leukemia Research Laboratories in the Nuffield Department of Pathology, University of Oxford.

David Mason was interested in developing a double immunostaining technique that can label  both human blood, and tissue antigens with alkaline phosphatase and peroxidase. The new method would  replace the horseradish peroxidase techniques,as the Immuno-peroxidase staining was not suitable for routine haematological diagnosis, the reason being that blood and bone marrow endogenous peroxidase could not be  fully inhibited. 

Alkaline phosphatase- anti alkaline phosphatase(APAAP), was a logical alternative; after making sure that neutrophils alkaline phosphatase could be stained without denaturing the cellular antigens. Moreover; blocking endogenous alkaline phosphatase enzyme activity was very essential to reduce nonspecific signals. 

Due to the high specificity of APAAP in staining B and T blood cell populations, and identify Leukemia lineage associated, specific antigens and stage differentiation. APAAP considered a major breakthrough in the field of hematology and in the diagnosis of Leukemia 

David Mason used monoclonal antibodies to increase the specificity and reduced the background of his  new assay.  The APAAP method allowed pathologists to store fixed blood sears and bone marrow samples for a long time without the loss of their cellular antigens. Moreover; APAAP  demonstrated its ability to stain both B and T blood populations and for that reason,  it was evaluated as a leukemia diagnosis method. In a study of over 250 malignant blood samples, APAAP identified lineage associated and lineage specific antigens, in addition to stage differentiation (e.g. CD34). 

APAAP hand many properties, that made it qualified for clinical diagnosis including: small size sample, easy to prepare, stain and store or read immediately, detect surface and cytoplasmic antigens, light microscope is sufficient for detecting abnormal cells, and can detect rare neoplastic cells e.g. CD30 positive Hodgkin cells or cytokeratin positive epithelial cells.

It was easy to distinguish between the bright red color of the antigen positive cell, and the blue haematoxylin nuclear staining and that was sufficient to identify cells by morphology and antigenic expression. This work was considered a major breakthrough in the field of hematology. 

Reviewer 2 Report

The commentary entitled “Immunocytochemical Labelling of Haematological Samples Using Monoclonal Antibodies” gave an interesting insight into the development of immune-alkaline phosphatase staining, performed on blood and bone marrow smears. The author emphasized that the APAAP staining method provided a simple and elegant technique, which could be applied in haematology laboratories to revolutionize leukaemia diagnosis. Meanwhile, the scientific achievements of David Y. Mason are highly acknowledged by the author.